# Methods for Studying *Magnaporthiopsis maydis*, the Maize Late Wilt Causal Agent

**Ofir Degani** [1,2,*] 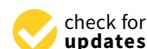**, Shlomit Dor** [1,2]**, Daniel Movshovitz** [1,2] **and Onn Rabinovitz** [2]

1   Faculty of Sciences, Tel-Hai College, Upper Galilee, Tel-Hai 12210, Israel; dorshlomit@gmail.com (S.D.); daniel.bobli@gmail.com (D.M.)
2   Department of Plant Sciences, Migal—Galilee Research Institute, Tarshish 2, Kiryat Shmona 11016, Israel; onnrab@gmail.com
*   Correspondence: d-ofir@bezeqint.net or ofird@telhai.ac.il; Tel.: +972-54-678-0114; Fax: +972-4-681-7410

**Abstract:** Late wilt, a destructive vascular disease of maize caused by the fungus *Magnaporthiopsis maydis*, is characterized by relatively fast wilting of maize plants closely before the physiological maturity stage. Previously, traditional microbiology-based methods have been used to isolate the pathogen and to characterize its traits. More recently, several molecular methods have been developed, enabling accurate and sensitive examination of the pathogen spread within the host. Here, we review the methods developed in the past 10 years in Israel, which include new or modified microbial and molecular techniques to identify, monitor, and study *M. maydis* in controlled environments and in the field. The assays inspected are exemplified with new findings and include microbial isolation methods, microscopic and PCR or qPCR identification, spore germination evaluation, root pathogenicity assay, *M. maydis* hyphae or filtrate effects on grain germination and sprout development, and a field assay. These diagnostic protocols enable rapid and reliable detection and identification of the pathogen in plants and seeds and studying the pathogenesis of *M. maydis* in susceptible and relatively resistant maize cultivars in a contaminated field. Moreover, these techniques are important for studying the population structure, and for future development of new strategies to restrict the disease's outburst and spread.

**Keywords:** *Acremonium maydis*; black bundle disease; *Cephalosporium maydis*; diagnosis; fungus; *Harpophora maydis*; late wilt; *Magnaporthiopsis maydis*; methods; real-time PCR

## 1. Introduction

Late wilt, or black bundle disease, is a vascular wilt disease of *Zea mays* L. (corn, maize) caused by the soil-borne and seed-borne fungus *Magnaporthiopsis maydis* (Samra, Sabet, and Hing; Klaubauf, Lebrun, and Crou [1]), with the synonyms *Harpophora maydis*, *Acremonium maydis*, and *Cephalosporium maydis* (Samra, Sabet, and Hingorani). Gams (2000) introduced the genus *Harpophora*, based on *Harpophora radiciola* for a group of species that are phialophora-like in morphology, with cylindrical, curved conidia [2]. The nuclear ribosomal internal transcribed spacer (ITS) phylogeny generated by Ward and Bateman (1999) [3] and Yuan et al. (2010) [4] showed that species of *Harpophora* were close to or grouped with *Gaeumannomyces* Arx and D.L. Olivier. Saleh and Leslie (2004) [5] also reported that *M. maydis* belonged to the *Gaeumannomyces–Harpophora* species complex, based on ITS, β-tubulin, and histone H3 gene sequences. Based on a two-locus phylogeny, the partial large subunit (28S) of the nuclear ribosomal RNA (nrRNA) gene operon (LSU) and the partial RNA polymerase II largest subunit gene (RPB1), Klaubauf et al. (2014) [1] recently transferred *M. maydis* to the genus *Magnaporthiopsis* J. Luo and N. Zhang, and treated *Harpophora zeicola* as a later synonym of *H. radicicola*.

The fungus reproduces asexually, and no perfect stage has been identified [5]. To date, late wilt has been reported in about 10 countries, with significant economic losses in Egypt [6], India [7], Spain and Portugal [8], and Israel [9]. In Israel, the sweet maize growth area covered about 2800 ha and yielded almost 57,000 metric tons of grains (data from the Israel Organization of Crops and Vegetables, 2017). Late wilt is considered to be the most destructive disease in maize-growing areas in this country, with up to 100% infection and total yield loss reported in some fields [10]. In Egypt, the cultivated maize area covered about 880,000 ha and yielded almost 7.2 million metric tons of grains [11], with a degree of loss that may reach up to 80% in infested fields [12]. The Egyptian, Indian, and Hungarian isolates of *M. maydis* differ in morphology, pathogenicity, and route of infection [13]. For example, in Egypt, four clonal lineages of *M. maydis* isolates show diversity in amplification fragment length polymorphism (AFLP), colonization ability, and virulence on maize [14–18]. In Spain and southern Portugal, 14 isolates of *M. maydis* were analyzed by inoculating maize-susceptible cultivars [19]. One of the isolates was more aggressive, caused significant weight reductions of both roots and above-ground parts.

Late wilt disease is characterized by relatively burst-wilting symptoms of maize plants, typically two weeks after the R1 fertilization growth stage (70 days after sowing) at the R3 growth stage (corn development described by [20]). The pathogen can negatively affect seedling emergence and cause seedling root necrosis [21,22]. First wilt symptoms appear near the flowering stage (from the R1 silking to the R2 blister stages), approximately 50–60 days after seeding [23] close to the male flowering stage. With disease progression, the lower stem dries out (particularly at the internodes) and has a shrunken and hollow appearance, with dark yellow to brownish macerated pith and brownish-black vascular bundles [24]. Late wilt is often associated with infection by secondary invaders, causing the stem symptoms to become more severe [25,26]. Fewer ears are produced, and kernels that form are poorly developed [9] and may be infested with the pathogen. Seed quantity [27] and quality [28] are correlated negatively to disease severity. The pathogen reacts differently to various environmental conditions [29,30] and is considered to be a poor competitor to other micro-organisms in the soil [23]. Spreading is through the movement of infested soil, crop residue [31], or seed-borne inoculum [32]. Lately, the former reports on potential secondary plant hosts were confirmed for *M. maydis*, *Lupinus termis* (lupine) [33], and cotton [22], and new hosts—watermelon and *Setaria viridis* (common name green foxtail)—were introduced [34].

Some agricultural [30,35], biological [11,36–38], and chemical controls [10,24,39,40] were able to reduce the pathogen's impact on commercial production. A combined treatment of fungicide seed coating and drip irrigation was recently demonstrated to be a cost-effective and powerful treatment to protect sensitive maize hybrids, even in heavily infested soils [28]. However, at present in Israel, late wilt disease is controlled by a more traditional management approach through the development of genetically resistant corn lines. Indeed, both in Israel (Onn Rabinovitz, Ministry of Agriculture, Consultation Service (Shaham), Beit-Dagan, Israel, unpublished data) and Egypt [12], great efforts are being devoted to increase maize productivity having a high resistance to late wilt. However, a review of the literature reveals that limited information is available on the pathogen's development in those apparently asymptomatic maize varieties and on the inheritance of resistance [41]. It was demonstrated that the pathogen also spread in relatively resistant plants that showed no symptoms [9], and that seeds of these apparently healthy, relatively resistant plants may therefore also spread the disease. Moreover, it appears that the resistance is sometimes restricted to a limited time period. Eventually, some sweet maize hybrids that show no signs of disease at the harvesting stage (typically three to four weeks after fertilization at the R3–R4 growth stages) will wilt and collapse after an additional one or two weeks [42]. To support this, it was shown in a field trial that, on day 72, the *M. maydis* DNA in the roots, stems, and leaves of the relatively resistant Royalty cultivar was similar to the fungal DNA spreading in the sensitive Jubilee cultivar on day 57 [9].

Previously, species-specific PCR primers capable of distinguishing *M. maydis* from other species in the *Gaeumannomyces–Harpophora* complex were developed [5] and are being used as a

diagnostic assay to track the pathogen spread inside the host plants in an infested field in northern Israel [9]. Later, a specific real-time PCR (qPCR) method was developed for the same purpose [10]. These molecular methods, together with symptoms evaluation methods, were used to study the effect of selected fungicides on the pathogen in a series of trials that would gradually increase the investment in time and work without wasting efforts if the treatment had not successfully passed the early steps.

The series of experiments was started in a Petri dish plate assay, followed by an in vitro seed assay (seed inoculation in an Erlenmeyer flask), a detached root assay, a seedling pathogenicity assay (sprouts up to the age of six weeks) in a growth chamber, potted plant evaluation over a full growth period (from seed to seed) in a greenhouse, and eventually a field assay [10,24]. The preliminary steps, up to the greenhouse trial, should be addressed in a critical manner, but they have several advantages. Despite the inconsistency of the ability of these assays in predicting results in the field [24,43], the preliminary steps are important to enhance prediction ability, ruling out ineffective treatments and choosing the ones that would most likely succeed in the field. In addition, the seedling pathogenicity assay is an important means for studying the early stages of the disease, developing new resistance maize strains and evaluating different ways of controlling maize late wilt.

Continued development of these preliminary steps and new methods for detecting and studying *M. maydis* are necessary. A scientific program aimed at developing disease control cannot be based solely on field trials during the growing season due to the long waiting time required until results are received, the great efforts involved in such experiments, and the changing environmental conditions causing inconsistency in the results. The current work aimed at summarizing the work methods to isolate, identify, study, and characterize *M. maydis* behavior and virulence accumulated in the past 10 years in Israel, together with new results demonstrating the use of these methods and expanding our knowledge of this phytoparasitic fungus.

## 2. Materials and Methods

### 2.1. Fungal Isolates and Growth Conditions

One representative isolate of *M. maydis*, called *Hm*2 (CBS 133,165), was selected for this study from our isolates library. This isolate, which was used throughout the work, is currently deposited at the CBS-KNAW Fungal Biodiversity Center, Utrecht, The Netherlands. Like the other isolates of *M. maydis* in this collection, this *M. maydis* strain was recovered from wilting maize plants sampled from a maize field in Sde Nehemia in the Hula Valley of the Upper Galilee (northern Israel) in 2001. Other Israeli *M. maydis* isolates that were included in this work were recovered from the sensitive Jubilee cv. and the relatively late-wilt-resistant Royalty cv. (both sweet maize cultivars from Pop Vriend Seeds B.V., Andijk, The Netherlands, supplied by Eden Seeds, Reut, Israel). All these isolates were characterized by their pathogenicity, physiology, colony morphology, and microscopic traits [9,29]. The morphological and microscopic characteristics of the isolates were identical to those of previously described strains found in Egypt and India [7,26]. Final confirmation was achieved by PCR-based DNA analysis [9,24]. All colonies were grown on potato dextrose agar (PDA) (Difco, Detroit, MI, USA) at 28 ± 1 °C in complete darkness.

### 2.2. Magnaporthiopsis maydis *Isolation*

Isolation of the pathogen from naturally infected plants was conducted as follows: internodes of symptomatic plants were sterilized in 5% sodium hypochlorite and split with a sterile knife; a small piece of discolored (from white to dark yellow or brown) vascular bundle was placed on PDA media. Alternatively, we used PDA with Hygromycin B (CAS No.: 31282-04-9, Sigma-Aldrich, Rehovot, Israel), at different concentrations, as will be described below in the Results. As noted by Saleh et al. [15], recovery of *M. maydis* is difficult, even from heavily infested material, due to its slow growth and the relative abundance of other, more rapidly growing fungi, particularly *Fusarium* spp. Cultures were

propagated from single conidia, and each isolate was identified as *M. maydis* according to cultural and microscopic characteristics [9,26] and molecular traits [24], as detailed below.

### 2.3. Magnaporthiopsis maydis *Detection and Identification*

*Microscopic characterization.* *H maydis* isolates were characterized using the slide culture technique [44]. A block of inoculated PDA agar sandwiched between two sterile cover glasses was placed in a sterile plastic Petri dish containing water agar and kept in an incubator at $28 \pm 1\,°C$ in complete darkness. After 2–6 days of growth, the slide culture was disassembled, and the appearance of hyphae structures, conidiophores, spores, and sclerotia bodies was observed and photographed using a light microscope equipped with a Moticam 5 (Motic Instruments, Richmond, Canada) microscope camera.

*DNA extraction.* Total DNA preparations were obtained from tissue samples of axenically grown maize tissue and from maize tissue known to be infected with *M. maydis* using the Extract-N-amp plant PCR kit (Sigma, Rehovot, Israel) according to the manufacturer's instructions, or using an alternative method according to the procedure of [45] with slight modifications [28]. According to the modified Murray and Thompson procedure, after grinding the tissue with 4 mL cetyltriammonium bromide (CTAB) buffer (0.7 M NaC1, 1% CTAB, 50 mM Tris-HC1 pH 8.8, 10 mM EDTA, and 1% 2-mercaptoethanol), 1.2 mL from the mixture was incubated for 20 min at $65\,°C$. The samples were then centrifuged at 14,000 rpm for 5 min at room temperature ($24\,°C$). The upper phase of the lysate (usually 700 μL) was then extracted with an equal volume of chloroform/isoamyl-alcohol (24:1). After mixing by vortex, the blend was centrifuged again at 14,000 rpm for 5 min at room temperature. This stage of chloroform/isoamyl-alcohol extraction was repeated twice. The supernatant (usually 300 μL) was then separated to a new Eppendorf tube and mixed with cold isopropanol (2:3). The DNA solution was mixed gently by inverting the tube several times, kept at $-20\,°C$ for 20–60 min, and centrifuged (14,000 rpm at $4\,°C$ for 20 min). The precipitate DNA was isolated and resuspended in 0.5 mL 70% ethanol. After another centrifugation (14,000 rpm at $4\,°C$ for 10 min), the precipitate DNA was isolated and left to dry in a sterile hood overnight. Finally, the DNA was suspended in 100 μl HPLC-grade water and kept at $-20\,°C$ until use.

*PCR-based method.* PCR was performed to amplify a specific *M. maydis* segment [5,9] with a Rapidcycler (Idaho Technology, Salt Lake City, UT, USA). Negative controls (in which water replaced the DNA template) were used to ensure the absence of primer dimers and contamination. The A200a primer set (Table 1) amplified a specific *M. maydis* 200 bp piece from an AFLP-derived species-specific fragment that was initially identified by [5]. The Am42/43 primer set (Table 1) amplified eukaryotic ribosomal DNA (18S rRNA gene product, rDNA) [46] and was used for a positive amplification control. Reaction mixtures were contained in a total volume of 20 μL: 1 μL of each primer (20 μM of each primer), 4 μL Red Load Taq Master (Larova, Teltow, Germany), 3 μL DNA sample, and 11 μL sterile double-distilled water (DDW). Cycling conditions for all primer pairs were $94\,°C$ for 2 min, followed by 35 cycles of $94\,°C$ for 30 s, $55\,°C$ for 30 s, $72\,°C$ for 1 min, and a final step of $72\,°C$ for 5 min. After the PCR, a 200 bp amplified DNA band was identified by electrophoresis on a 1.5% agarose gel (Lonza, Rockland, ME, USA). The original (old) primers [9] were selected for the PCR analysis done throughout this work.

**Table 1.** Primers for *Magnaporthiopsis maydis* detection.

| Pairs | Primer | Sequence | Uses | Amplification | References |
|---|---|---|---|---|---|
| Pair 1 (old primers) | A200a-for<br>A200a-rev | 5′-CCGACGCCTAAAATACAGGA-3′<br>5′-GGGCTTTTTAGGGCCTTTTT-3′ | PCR and qPCR | *M. maydis* AFLP-derived specific fragment | [9] |
| Pair 2 (new) primers) | A200a-for<br>A200a-rev | 5′-CCTAGTAGTCCCGACTGTTAGG-3′<br>5′-TTGGTTCACCGTCTTTTGTAGG-3′ | PCR | *M. maydis* AFLP-derived specific fragment | [24] |
| Pair 3 | Am42<br>Am43 | 5′-CAACTACGAGCTTTTTAACTGC-3′<br>5′-CAAATTACCCAATCCCGACAC-3′ | PCR Control | Eukaryotic ribosomal DNA 18S rRNA gene product, rDNA | [46] |
| Pair 4 | Cox-F<br>Cox-R | 5′-GTATGCCACGTCGCATTCCAGA-3′<br>5′-CAACTACGGATATATAAGRRCCRRAACTG-3′ [1] | qPCR Control | cytochrome c oxidase (COX) gene product | [47] |

[1] The R symbol represents Guanine or Adenine (purine). The synthesized primer contained a mixture of primers with both nucleotides.

*qPCR-based method*. All quantitative real-time PCR (qPCR) reactions were performed, as previously described [10], using the ABI PRISM® 7900 HT Sequence Detection System (Applied Biosystems, CA, USA) for 384-well plates. Plant tissue samples (root and stem) from each experiment were analyzed separately by qPCR. The A200a primers were used for qPCR (sequences in Table 1). The gene coding for the last enzyme in the respiratory electron transport chain of the eukaryotic mitochondria—cytochrome c oxidase (*COX*)—was used as a "housekeeping" reference gene to normalize the amount of DNA [47]. This gene was amplified using the *COX* F/R primer set (Table 1). The qPCR reaction mixture was as follows: 5 µl total reaction volume was used per sample well—2 µL of sample DNA extract, 2.5 µL iTaq™ Universal SYBR® Green Supermix (Bio-Rad Laboratories Ltd., Rishon Le Zion, Israel), 0.25 µL forward primer, and 0.25 µL reverse primer (10 µM from each primer to a well). The qPCR conditions were as follows: precycle activation stage, 1 min at 95°C, 40 cycles of denaturation (15 s at 95 °C), annealing and extension (30 s at 60 °C), followed by melting curve analysis. Relative gene expression was calculated according to the ΔΔCt model [48]. Efficiency was assumed to be the same for all samples. All amplifications were performed in triplicate.

### 2.4. Magnaporthiopsis maydis *Growth Behavior Tests*

The growth tests are an important means for studying *M. maydis* behavior in response to various ambient conditions. These conditions may include physical environmental stresses, interactions with other micro-organisms, plant extracts, soil solutions, nutrients, growth regulators, or antifungal compounds. These challenging environmental conditions can be tested by adding the inspected solutions to spores or to detached roots, as will be detailed below.

*Spore germination assay*. This assay was performed as previously described [29]. To induce sporulation, cultures were grown at 28 ± 1 °C in a humid atmosphere on the surface of PDA. After four days, spores were harvested, washed, and scraped off the agar surface with 1 mL sterile deionized water. The spore suspension was kept at 4 °C for up to one day. The spores were diluted to approximately 40 spores per µL in DDW before use. The spores' suspensions in Eppendorf tubes were then incubated in a rotary shaker at 150 rpm at 28 ± 1 °C in the dark. The percentage of germinating conidia was determined after incubation for 0, 3, 6, 8, or 16 h by direct counting in 2 µL drops on a glass slide using a light microscope equipped with a Moticam 5 microscope camera. A representative photo of each treatment is presented. The criterion for germination was the observation of any germ tube emerging from the examined spores. Each assay was performed in six independent replications, and the entire experiment was repeated twice.

*Detached root assay*. The root assay (reported earlier by [49]) provides us with a more realistic evaluation method than the culture media plates since it inspects fungal development on its natural host. Plants were grown under the conditions described below in the seedling assay protocol. For the selection of roots, care was taken to collect roots of similar diameter and pigmentation. Lateral, young, and white roots, about 2 cm long, were removed from potted 20-day-old maize seedlings (Jubilee cv. or Royalty cv.) and washed under tap water to remove the soil. The roots were disinfected using 70% ethanol solution and dried in a pre-sterilized fume hood. Each root was then placed individually in a sterile plastic Petri dish lined with Whatman 3 mm filter paper. Ten milliliters of autoclaved DDW was used for wetting the filter paper beneath the infected detached roots in a Petri dish. A 6-mm-diameter culture PDA agar disk taken from the growing edge of a 4–6-day-old fungal colony (grown at 28 ± 1 °C in the dark) was placed at the cut end of each root. Negative control roots were left untouched. There was no difference between untreated roots or roots treated with a sterile 6-mm-diameter PDA agar disk (thus data are shown only for the untreated roots). The Petri dishes were sealed with Parafilm (a plastic paraffin film) and incubated at 28 ± 1 °C in the dark. Each treatment was done in two independent repeats and the results were similar; data for one representative root of each treatment are shown. The entire experiment was repeated twice. The lengths of root infection threads (seen as a dark filament within the root) were identified, measured, and photographed 3, 6, or 10 days

after inoculation. For DNA extraction, one segment was cut at 1 cm from the infected or uninfected root cut ends. DNA isolation and PCR were conducted as described above.

*2.5.* Magnaporthiopsis maydis *Virulence Assays under Controlled Conditions*

*M. maydis* pathogenic behavior was tested in a series of experiments starting from in vitro seed trials in an Erlenmeyer flask, followed by a seedling assay (sprouts up to the age of six weeks) in a growth chamber. These steps are important to determine the *M. maydis* isolates virulence degree and for fast evaluation of antifungal compounds to restrict the pathogen. Success in the controlled-condition virulence assays is an important phase before final evaluation in a field trial. These inspections steps will be described in detail below.

In vitro *seed infection*. Sweet maize, Jubilee cv. and Prelude cv. (from SRS snowy river seeds, Australia, supplied by Green 2000 Ltd., Bitan Aharon, Israel) were chosen for the seed pathogenicity test due to their high susceptibility to late wilt disease [9,10]. To examine the ability of the pathogen to infect maize seeds in vitro, we inoculated seeds by soaking them in a fungal suspension and then detecting the presence of the fungus in the inner tissues, as previously described [10]. Ten seeds of the selected maize cultivar were dipped in 1% (*v/v*) sodium hypochlorite for 3 min, washed in sterile DDW and then placed in a 250 mL Erlenmeyer flask with 20 mL autoclaved DDW. To each of the Erlenmeyer flasks containing seeds, we added three 6-mm-diameter agar disks cut from the margins of *M. maydis* colonies (grown previously on PDA at 28 °C in the dark for 4–6 days), and the Erlenmeyer flasks were incubated at 28 °C in the dark. Seeds germination percentage determination and wet weight measurements of all the seeds in each treatment were conducted one week after inoculation. A germinating seed was defined as a seed in which the seed coat was broken by the radicle. For wet weight assessment, sprouts were collected at the end of the experiment, dried gently with paper towels, and the fresh biomass was measured for each bud individually using analytical scales. Each treatment included six independent replications, and the entire experiment was conducted twice with similar results.

For molecular inspection, at the end of the experiment (after 7 days of inoculation), a sample of three seeds from each repeat in each treatment was washed thoroughly with a 70% ethanol solution and then with autoclaved DDW to remove any residuals of the fungus that may be attached to the seeds' surface. The seeds were ground to a powder using liquid nitrogen, and DNA was obtained as described above. The purified DNA was used as a template for the PCR or qPCR reaction.

Magnaporthiopsis maydis *exudates effect on seed germination*. To evaluate pathogenicity variation, we studied the effect of *M. maydis* isolates metabolites on maize seed germination according to [50]. The culture filtrate of our *M. maydis* isolate collection and the susceptible maize Prelude cultivar were used to perform the following tests. Five 6-mm-diameter agar disks cut from the margins of each *M. maydis* isolate colony (grown for 4–6 days at 28 ± 1 °C in complete darkness) were added to 150 mL sterile potato dextrose broth (PDB) in a 250 mL Erlenmeyer flask. Cultures were grown at 28 ± 1 °C in complete darkness on a rotary shaker at 150 rpm for 22 days. The liquid medium of each culture was filtrated through two Whatman No. 1 filter papers in a Büchner funnel and centrifuged at 6000 rpm for 20 min. After filtration through a 0.4 μm membrane filter, the filtrate obtained was immediately used for treating the seeds as follows: grains of Prelude maize cultivar were soaked in the isolates culture filtrate for 6 h; grains soaked in sterilized PDB liquid medium or in DDW were used as a check treatment. The treated seeds were then dried, moved to Petri dishes (10 grains/dish), and kept on water-moistened (with sterile tap water) Whatman No. 1 filter paper at 28 °C. Each treatment was conducted in triplicate. The number of germinating seeds was recorded after two days.

*Seedlings assay in a growth chamber*. Maize sprout virulence evaluation is another way of studying pathogenicity variations within the *M. maydis* population and was also performed here to evaluate several inoculation methods. This assay (described formerly [10,24]) was performed using the sensitive maize hybrids Jubilee cv. and Prelude cv. Three inoculation methods were compared: (1) the inoculum method consisting of naturally infested soil taken from a commercial maize field; (2) the direct

application of hyphae or culture disks to naturally infested soil or to a commercial disease-free soil mixture; and (3) adding sterilized inoculated barley seeds to the abovementioned soils.

For the direct application of hyphae inspection, the fungus was grown in PDB liquid medium for 3–5 days (at $28 \pm 1$ °C in the dark on a rotary shaker at 150 rpm), and 200 mg mycelium suspension was added directly to each seed. Alternatively, four agar disks (6-mm-diameter) from young (3–5-day-old) *M. maydis* colonies (grown at $28 \pm 1$ °C in the dark) were added to each seed. To perform this procedure with minimal interference in the seedling development, a glass tube (10 × 75 mm) was inserted next to each seed with seeding. On the inoculation day (7–10 days from sowing, when the plants first emerge above the ground surface), the glass tubes were removed, and the mycelium suspension or culture disks were added instead (approximately 4 cm beneath the ground surface). The remaining hollowed area was filled with soil. The control plants (non-inoculated plants) were grown under the same conditions.

The infected sterilized barley seeds soil inoculation protocol used was based on a predeveloped method [9]. These barley seeds were only used to spread the pathogen in the soil, as done previously with sorghum seeds [16]. The autoclave-sterilized seeds were soaked overnight in *M. maydis* inoculum suspension. A control treatment included barley seeds soaked in an equal volume of sterile water. For soil inoculation, the top 20 cm of the soil was removed from each pot and mixed with 20 g sterilized inoculated barley seeds and then returned to the pot. This procedure was done immediately before seeding. A control treatment included barley seeds soaked in an equal volume of sterile water.

The experiment was conducted in triplicate (data are shown only for the last experiment, but similar results were obtained for all three repetitions). Each treatment included 4–6 independent replications (pots), as will be detailed in the figure legend. Two to five maize seeds were sown in a 2.5 L pot about 4 cm beneath the surface. The soil mixture (Shacham Givat Ada, Givat Ada, Israel) was commercial and non-sterilized, composed of 65% coco, 20% peat, 10% tuff (4–10 mm volcanic stones) and 5% Multicote® Agri, a controlled-release fertilizer (Haifa Chemicals Ltd., Haifa, Israel) *w/w*. Alternatively, naturally infested soil taken from a commercial maize field (mixed with 30% Perlite No. 4 to enable higher soil water content and to prevent soil compaction) was used. Watering was done by adding 100–160 mL DDW every 72 h to the pots using a computerized drip irrigation system with a drip flow rate of 2 L/H (the exact amount of water was adjusted during the experiment to maintain sufficient ground moisture). When the field soil was used without the addition of Perlite, the watering amount was doubled to maintain equal soil moisture as the commercial soil mixture. All the plants used for the pathogenicity assay were grown in a growth chamber under a constant temperature of $28 \pm 3$ °C, relative humidity of 45–52%, with a 12 h photoperiod illuminated by cool-white fluorescent tubes (Philips, Eindhoven, The Netherlands).

For wet weight assessment, the seedlings were sampled (according to a timetable that will be described individually for each experiment in the figure legend), cleaned of visible soil by rinsing thoroughly under running tap water, and dried gently with paper towels. The root and shoot of each sprout were separated by a scalpel, and the fresh biomass of each part was measured individually using analytical scales. The aboveground height (from the first node to the shoot tip) of each plant was measured separately and the number of leaves counted.

To prepare the plants for DNA purification, the seedlings' underground parts were rinsed twice in sterile water for 30 s each time. Under a sterile biosafety hood, tissue samples were excised from the root and the near-surface hypocotyl tissues. Tissues were sampled by removing a cross-section of approximately 2 cm in length from each plant. Samples from the five plants of each pot were combined, and the total weight was adjusted to 0.4 g and considered to be one repeat. Tissue samples were placed in universal extraction bags (Bioreba AG, Reinach, Switzerland) with 4 mL CTAB buffer, and the tissue was ground with a hand tissue homogenizer (Bioreba, Switzerland) for 5 min until the tissues were completely homogenous and then used directly for the DNA isolation (see above).

*2.6. Magnaporthiopsis maydis Infested Field Experiments*

　　A field experiment was carried out to study the ability of *M. maydis* to infect, spread, and lead to symptoms outbreak in maize cultivars varying in late wilt susceptibility. The experiment was carried out during the spring and summer of 2016 and included two representative maize cultivars: susceptible Prelude cv. and relatively late-wilt-resistant Royalty cv. The pathogenesis of the two cultivars was evaluated using the traditional PCR and the new qPCR detection methods [10]. The experiment was performed in the southern area of a maize field (Mehogi 5 maize plot) near Kibbutz Amir in the Hula Valley (Shemesh field crops partnership, Upper Galilee, Northern Israel), which has been known to be infested with late wilt for many years. This area was part of a large maize field used for grain production; the rows nearby were planted with other maize hybrids and are not included in the current study. Plots were arranged in the field using a randomized complete block design. The area included 48 plots, each containing two rows (eight plots per treatment). Each row was 12 m long and contained 3.3 maize plants m$^{-1}$ (40 plants per row). Row spacing was 96.5 cm. In addition to the nontreated plots of the two cultivars, Prelude and Royalty (these plots were used here as a control), two seed treatments were introduced and inspected: (1) seeds were pretreated (in a common commercial procedure by Gadot Agro, Kidron, Israel) with Azoxystrobin, 0.0025 mg active ingredient per seed (within the standard concentration range used by Syngenta AG, Basel, Switzerland); and (2) seeds were pretreated with mycorrhizal fungi (Rootella S™ product from Groundwork BioAg, Mazor, Israel). The field was watered with a combination of a frontal irrigation system and a 20 mm drip irrigation line (Dripnet PC 1613F, Netafim, Fresno, CA, USA) for two rows (200 and 400 mm, respectively, per growing season in total).

　　Seeding was performed on 25 May 2016, and germination (with the frontal irrigation system) one day later. Plants emerged above the ground surface approximately six days after planting. Plants were first pollinated when they reached 70% silk, 50 days after sowing (DAS). The pollination continued for three days. Wilt determination was carried out 19 and 25 days after fertilization (DAF) (69 and 75 DAS) by calculating the percentage of the plants that show typical late wilt dehydration symptoms: color alternation of the upper leaves to light-silver and then to light-brown and rolling inward from the edges of the entire leaf. Yield determination included all of the upper part plant cobs in a 5-m-long section of each of the experimental rows. Until harvest day (27 DAF, 77 DAS), the sensitive Prelude cv. plots were collapsed, and total yield loss was recorded. In the relatively resistant maize Royalty, the symptoms were considerably less severe, enabling yield evaluation.

　　For molecular diagnosis, three plants were collected arbitrarily from the Prelude cv. and Royalty cv. plots at approximately 10-day intervals from sowing onwards. Sampling was made on days 13 (roots), 22 (roots), 32 (roots), 42 (roots), 53 (roots and stems), 62 (stems), and 72 (stems) after seeding. Different plant tissues (root or stem) were sterilized separately with 70% ethanol and then washed with autoclaved DDW. DNA was extracted and analyzed, as described above.

*2.7. Statistical Analyses*

　　When assessing the *M. maydis* infection outcome on symptoms in the in vitro seed inoculation, in the growth chamber sprout infection, or in diseased field plants, we used completely randomized statistical designs. Student's *t*-test or Tukey–Kramer post-hoc test following a significant one-way ANOVA result (with a significance threshold *of p* = 0.05) was used for comparisons of treatment means to the control.

## 3. Results

*3.1. Magnaporthiopsis maydis Isolation and Detection*

　　*Fungal isolation.* Isolating *M. maydis* directly from naturally infested soils may be a difficult task to perform since the pathogen is scattered in small quantities in the soil and the disease spreading is ununiformed in the field (an example is shown in Figure 1A, brown patches). Moreover, the relatively slow growth rate of *M. maydis* on agar plate media compared with other fast-growing microorganisms present in the soil samples, especially *Fusarium* spp., makes this goal even more difficult to achieve.

We made substantial efforts in isolating *M. maydis* from known infested soils (commercial fields in which late wilt caused severe damages), but with minor success (data not shown). Thus, to this end, we used Hygromycin-embedded growth media in a concentration that still allowed *M. maydis* to grow (up to 50 µg/mL). The pathogen was also isolated directly from symptomatic maize plants (as described in Materials and Methods, Figure 1B,C) using the Hygromycin–PDA media plates (Figure 1D). After transferring the *M. maydis* colony several times to a new plate to ensure that the culture contained a sole organism, the pathogen was identified by its morphological and molecular traits.

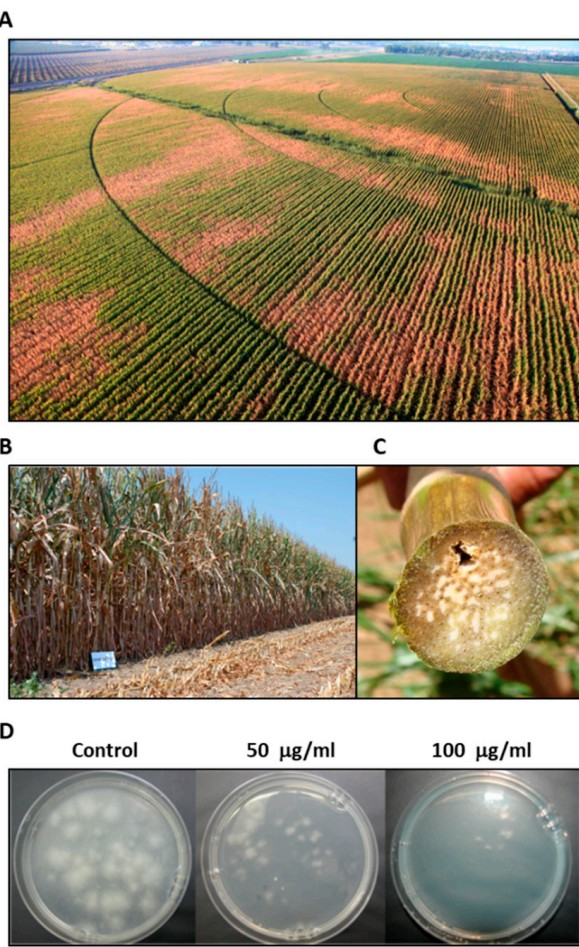

**Figure 1.** *Magnaporthiopsis maydis* isolation. (**A**) An aerial photograph of a late wilt diseased field (photographed by Asaf Solomon in the summer of 2017). This commercial field is located near Kibbutz Lohamei HaGeta'ot in northern Israel, and the semi-sensitive fodder maize (*Zea mays* L.) cultivar is Colossus from HSR Seeds, Orbost, Australia, supplied by CTS, Hod Hasharon, Israel. The brown patches are wilting plants infected by *M. maydis.* (**B**) A photograph of the Colossus cv. taken in a commercial maize field near Yavne (located in the southern coastal plain of Israel) in 2014. On harvest day 99 days after sowing (DAS), 48 days after fertilization (DAF), the plants were severely affected by late wilt disease, especially in the aboveground lower parts. Wilt symptoms include stem and leaf yellowing and dehydration. (**C**) Disease symptoms appearing on the stem in cross-section include rot and color alteration of the tissue to a yellow-brown hue and the appearance of a blocked vascular bundle with a brown substance seen as dark spots. (**D**) Colony agar plates with Hygromycin B-embedded growth media (potato dextrose agar, PDA) in a concentration that still allows *M. maydis* to grow (up to 50 µg/mL). The Control plate contains PDA without Hygromycin B. These plates were used to isolate *M. maydis* from infested soils and plants.

*Microscopic morphology*. Sporulation occurred within 2–4 days of transferring colonies to new plates. Conidiophores were hyaline, septate, and branched (Figure 2A–D), and usually carried 4–7 single-celled, oval to cylindrical, hyaline, smooth-walled conidia (Figure 2E,F). Conidia were produced abundantly and were oval in shape, about 5–7 µM wide and 10–15 µM long. Conidia germinated rapidly, usually starting with one bipolar germ tube; however, while developing, one to three germ tubes were formed (this will be demonstrated in the spore germination assay below). Hyphae were 2–4 µM in width, hyaline, septate, branched, and decumbent. Anastomosis of germ tubes often occurred (Figure 2G). Uniquely, the hyphae tended to bend and wrap around themselves to form typical coil shapes (Figure 2H,I). A compact mass of hardened fungal mycelium and aggregates of dark-colored, thick-walled cells typically started to form in old colonies after 2–3 weeks of incubation (Figure 2J–M). These aggregates will eventually develop to form sclerotium-like bodies enabling the fungus to survive under extreme environmental conditions.

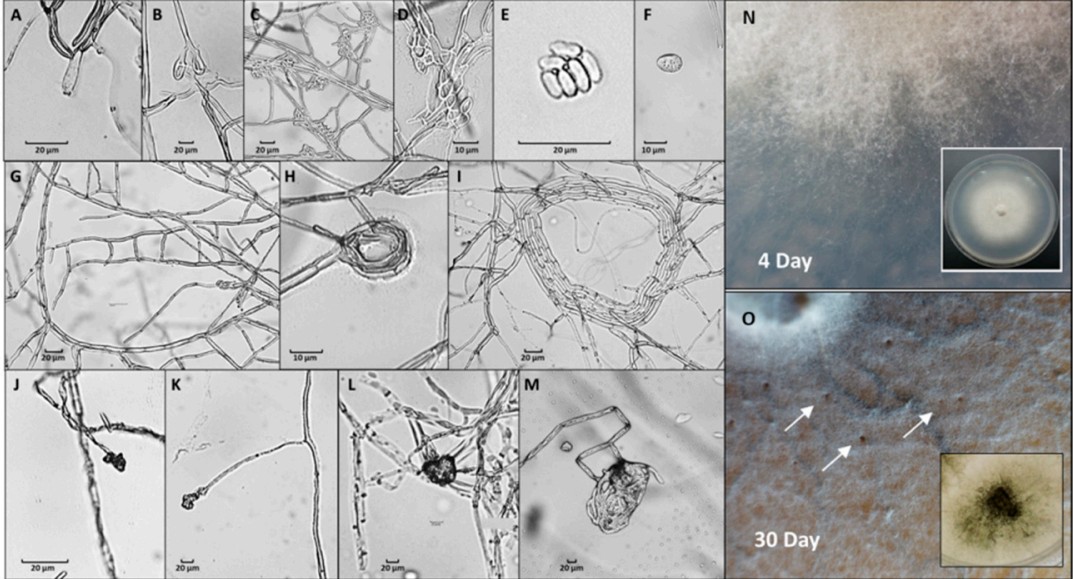

**Figure 2.** Morphological and microscopic characterization of *Magnaporthiopsis maydis*. (**A–D**) Conidiophores. (**E,F**) Conidia. (**G–I**) Typical hyphae structures, including anastomosis of germ tubes (**G**) and coil shapes (**H,I**). (**J–M**) Sclerotium-like bodies formed in old cultures (usually 30 days old or more) seen as a compact mass of hardened fungal mycelium and aggregates of dark-colored, thick-walled cells. (**N**) White young colony (4 days old on PDA). (**O**) A dark mature colony (30 days old) containing typical sclerotium-like bodies appears as dark bumps (approximately 0.1 mm in diameter) on the colony's surface (marked by white arrows). The old colonies exhibiting a "rhizoidal" or "hyphal rope" appearance ((**O**), insert). All cultures were grown at 28 °C in the dark.

*Cultures morphology*. Colonies were flat and white at first (Figure 2N), then became gray to black in old cultures, and formed dense aerial mycelium on PDA (Figure 2O). Dark-grown colonies covered a 90 mm plate within 6–7 days. The margins of the older (10–21-day-old) colonies had rope-like strands formed by wavy hyphae (Figure 2O, insert) and contained typical sclerotium-like bodies appearing as dark bumps (approximately 0.1 mm in diameter) on the colony's surface (Figure 2O, arrows).

*DNA-based identification*. Traditional molecular PCR detection of the pathogen is based on AFLP amplification of a specific *M. maydis* 200 bp piece fragment (Figure 3). All of these primer sets could efficiently detect *M. maydis* DNA inside the host plant tissues; however, conventional PCR was not sufficiently sensitive to detect low levels of *M. maydis* DNA in plant tissues [24]. In fact, the PCR tracking method in field assays was only able to detect the pathogen DNA from day 40 after seeding onwards [9]. Therefore, a more sensitive qPCR-based method will be presented in the continuation of the text.

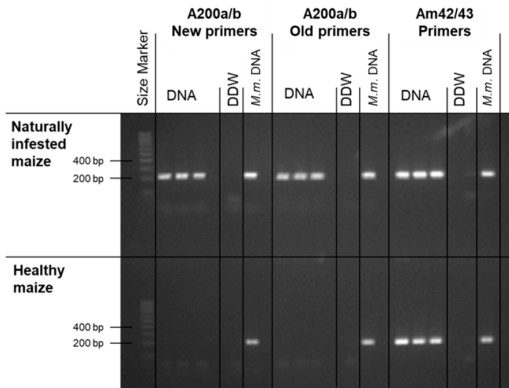

**Figure 3.** PCR diagnosis of late wilt in maize plant samples. Stem tissue of naturally *Magnaporthiopsis maydis*-infested sensitive Jubilee maize cv. (approximately 60 days old) was sampled from commercial field plants, in the Hula Valley in the Upper Galilee in northern Israel. The control healthy maize tissue was taken from roots of healthy Jubilee cv. sprouts (up to 40 days) grown under controlled conditions. The samples were inspected using PCR amplification of the unique *M. maydis* oligonucleotide (A200a/b primers) and rDNA—18S eukaryotic ribosomal DNA (Am42/43 primers). Electrophoresis results show a 200 bp DNA band unique to the pathogen. Primers are detailed in Table 1. Size marker—100 bp DNA ladder, New England Biolabs (NEB), Ipswich, Massachusetts, USA, supplied by Ornat, Rehovot, Israel; DDW: distilled and deionized water negative control, used as a template in the PCR mixture to ensure absence of DNA contamination; and *M.m.* DNA: DNA extracted from a *Hm*2 isolate colony grown on an agar rich medium plate was used as a positive control.

*3.2. Magnaporthiopsis maydis Growth Behavior Tests*

*Spore germination assay.* During pathogenesis, *M. maydis* spores are present in the host plant xylem vessels [26] and thus may be subjected to the influence of the host environment. The spore germination assay may serve as an important research tool for investigating the influence of various environmental and host conditions on pathogen development and spread [29,49]. This assay should be evaluated based on the plate's sensitivity assay (Figure 1D) and the detached root pathogenicity assay that will be presented below. We calibrated this assay to determine the preferred time schedule recommended for use (Figure 4). The spore germination curve has a lag period in the first six hours of incubation, so the linear development phase is around nine hours of incubation. This is probably the best incubation time for identifying small variations in the growth rhythm. A longer incubation period will result in a decrease in the sensitivity of this test.

*Detached root assay.* The effect of environmental suppressors, enhancers, or regulators on the ability of *M. maydis* to infect detached roots can be assessed efficiently using the detached root assay when the fungus grows on its natural food source. Detaching the roots may accelerate their senescence and susceptibility to the pathogen (as shown in leaves for the foliar pathogen *Cochliobolus heterostrophus* [51]). This is important for increasing the sensitivity of this test. The roots themselves may be subject to the plants' hormonal influence that could alter their susceptibility to pathogen invasion [52]. To calibrate this assay, we used young, white, and partly transparent lateral roots from two sweet maize cultivars, the sensitive Jubilee cultivar, and the relatively resistant Royalty cultivar. Figure 5 shows the results obtained after three and six days of incubation for the Jubilee cv. and after 10 days for the Royalty cv. The assay was conducted with the roots lined in fresh water under optimal temperature (28 ± 1 °C) and light conditions (totally dark conditions). Similar results were obtained for both maize cultivars regardless of the differences in their susceptibility to the pathogen. Dark brown root infection threads (seen as dark filaments within the root) were clearly distinguishable from the remaining healthy root, whereas the non-inoculated (uninfected) negative control root remained clear as expected (Figure 5A). First results were obtained after three days when the inoculated roots started to develop a short infection thread (11.1 mm). The development of the infection thread continued with an average elongation rate of 3–4 mm/day. Disruption in this optimal rhythm can be measured easily, as previously reported [49]. Molecular tracking of a sample segment cut 1

cm from the inoculated end of the Jubilee roots on day six confirmed that this thread was the consequence of the presence of *M. maydis* and not of unwanted other contaminants (Figure 5B).

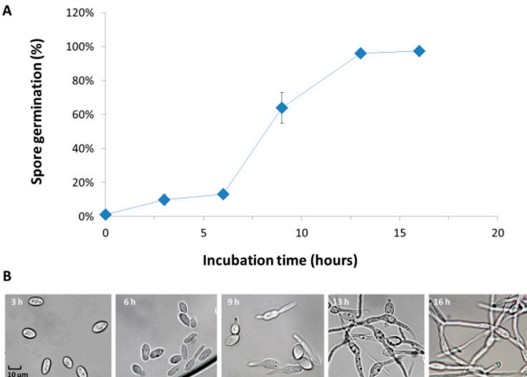

**Figure 4.** *Magnaporthiopsis maydis* Spore germination curve. Spores were washed from the growth media surface of 4-day-old colonies grown at 28 ± 1 °C in a humid atmosphere in the dark. The spores' suspensions in Eppendorf tubes were then incubated in a rotary shaker (150 rpm) under the same conditions for 0, 3, 6, 8, or 16 h. A. The percentage of germinating conidia (with any visible germ tubes emerging) was determined after incubation by direct counting using a light microscope. Values represent the average of five replicates. Error bars indicate standard error. B. A representative photograph of the spore solutions at the examined time interval.

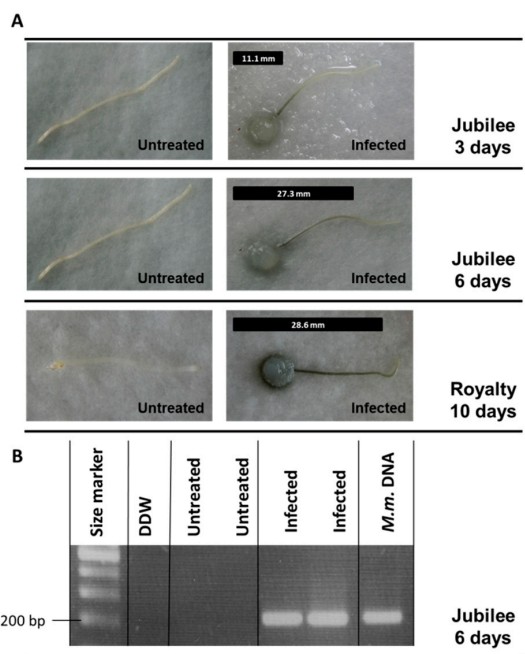

**Figure 5.** Detached root pathogenicity assay. Young and white lateral roots, about 2 cm long, were removed from potted 20-day-old maize seedlings (the susceptible Prelude cv. and the relatively resistant Royalty cv.) and inoculated by placing a 6 mm diameter *Magnaporthiopsis maydis* culture agar disk taken from the margins of a 4–6-day-old fungal colony (grown at 28 ± 1 °C in the dark) on the cut end of each root. The inoculated roots were placed separately in Petri dishes containing DDW and incubated in moist at 28 ± 1 °C in the dark. Negative controls are roots without pathogen inoculation under the same conditions. A. Progression of the pathogen infection thread inside the xylem tissue of each root (seen as a dark filament within the root) was evaluated quantitatively at the indicated inoculation times and marked in the photograph by a black line positioned above each root. B. In order to identify the fungus DNA in the above-treated root tissues, one segment was cut at 1 cm from the infected end of each root six days after inoculation. DNA isolation and PCR were conducted to determine the presence of the pathogen using a PCR-based method, amplified at 200 bp *M. maydis*-specific oligonucleotide. The PCR abbreviations are described in Figure 3.

### 3.3. Magnaporthiopsis maydis *Virulence Assays*

In vitro *seed infection.* In this study, we tested the pathogen's ability to infect seeds in vitro and under controlled conditions in Erlenmeyer flasks. Inoculating the seeds with *M. maydis* significantly ($p < 0.05$) suppressed the sensitive Jubilee seed germination compared to the control (Figure 6A). The seed inoculation also had a negative effect on the growth of this maize cultivar, expressed as lower fresh biomass (with a significant difference from the control of $p < 0.05$, Figure 6B). In comparison, the addition of *M. maydis* to seeds of the relatively resistant Royalty cv. resulted in only a minor (and insignificant) influence on germination and biomass measures. The fungal addition to seeds of both maize cultivars led to PCR detection of the pathogen inside the infected seeds (Figure 6C). However, the band intensity in the electrophoresis gel, visualized after the PCR amplification, was very weak, and thus revealed the limitation of this conventional molecular technique.

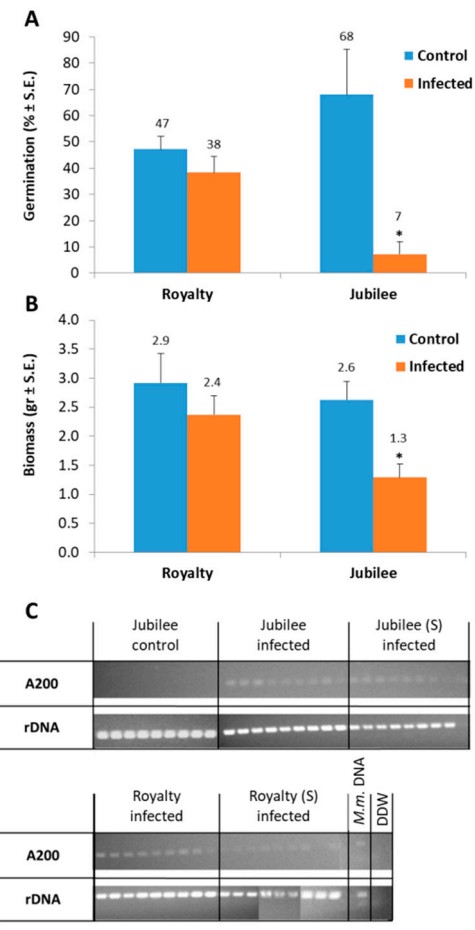

**Figure 6.** In vitro seed infection. Seed germination percentage (**A**) and wet weight (**B**) of the susceptible maize genotype Jubilee cv. and the relatively resistant Royalty cv. upon inoculation with an *Hm2* isolate of *Magnaporthiopsis maydis* grown in Erlenmeyer flasks (in vitro) for one week. Untreated seeds (seeds soaked only in water) were used as negative controls. Vertical upper bars represent the standard error of the mean of six replications (Erlenmeyer flasks, each containing 10 seeds). When existing, significance from the control (untreated) is indicated as * = $p < 0.05$. Electrophoresis results (**C**) show PCR amplification of the unique *M. maydis* oligonucleotide (A200, upper panel) and rDNA (18S eukaryotic ribosomal DNA, lower panel). A sample of three seeds from each repeat in each treatment used to extract the DNA for the PCR. Jubilee or Royalty (S) infected – autoclaved sterilized seeds that were infected, used as positive control treatment. The other PCR abbreviations are described in Figure 3.

Magnaporthiopsis maydis *exudates effect on seed germination.* Here, we investigated whether a filtrate of *M. maydis* grown on rich liquid medium could affect the susceptible maize cultivar Prelude

seed's germination. Soaking the seeds in each of our *M. maydis* isolates library culture filtrates in Petri dishes for two days resulted in germination inhibition. Interestingly, this inhibition outcome varied among the isolates; therefore, it may serve as one of the measures that could assist in determining the degree of *M. maydis* isolates virulence. To elaborate on this, soaking the maize grains in the fungal filtrate before seeding inhibited their germination up to 40% (in the *Hm7* isolate) compared to those soaked in blank water (Figure 7A). Among the isolates were some that only negligibly suppressed, or did not suppress at all, the germination (for example, the *Hm8* isolate). Moreover, this reduction in seed germination (after *M. maydis* filtrate pre-incubation) was only a consequence of delayed germination and not the result of seed vitality loss (Figure 7B).

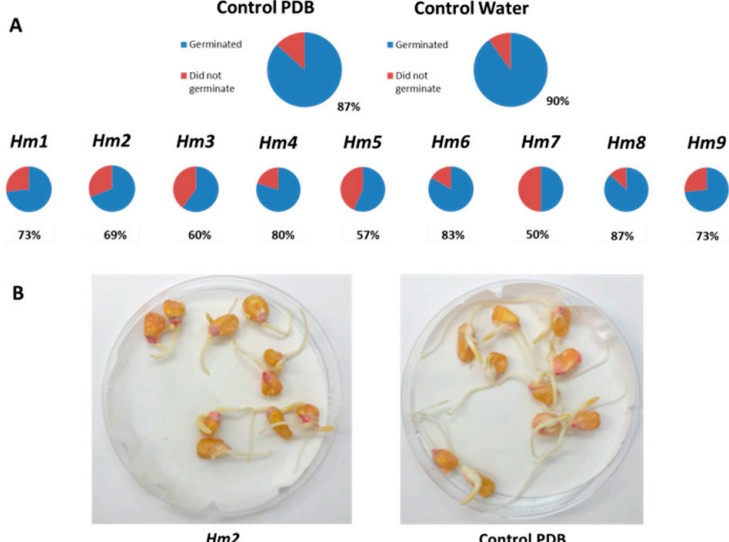

**Figure 7.** Effect of *Magnaporthiopsis maydis* culture filtrate on seed germination. (**A**) Seeds of Prelude cv. were soaked in the different isolates culture filtrate for 6 h. Grains soaked in sterilized PDB (potato dextrose broth) liquid medium (left) or in DDW (right) were used as control treatments. The treated seeds were then dried, moved to Petri dishes (10 grains/dish), and kept in water-moistened filter paper at 28 °C. Each treatment was conducted in triplicate. Numbers of germinating seeds were recorded after two days. (**B**) A representative photograph of isolate *Hm2* and the PDB control.

*Seedling assay in a growth chamber.* The evaluation of a qPCR assay compared to a conventional PCR assay for detecting and monitoring *M. maydis* DNA inside the host tissues is another purpose of the current work. A test for pathogenesis in sprouts (up to 40 days, four-leaves stage) in a growth chamber had been previously developed in our lab [24] and was used here (Figures 8–11) together with the older PCR and the newer qPCR detection to identify a preferred infection method.

Growing sensitive maize cultivars, such as Jubilee, on naturally infested or deliberately infected soil, resulted, just like in the in vitro seed assay, in suppressing phenological development measured in a decreased biomass of the sprouts. Naturally infested soil had a stronger and statistically significant ($p < 0.05$) inhibition effect on seedling growth compared to the artificially infected soil (Figure 8). Following the development of this pathogenicity inspection technique, we used this procedure to evaluate three inoculation techniques for the seedling pathogenicity experiments. The techniques included: adding four culture agar disks (6 mm in diameter) or 200 mg hyphae suspension directly to each seed 7–10 days after the sowing (when the sprouts emerged above the ground surface), or mixing the top 20 cm of the soil immediately before seeding with sterilized barley seeds soaked previously overnight in *M. maydis* mycelial suspension (Figure 9). These barley seeds were only used to spread the pathogen in the soil, as was done previously with sorghum seeds [16]. After the growing period in a growth chamber, and as early as six weeks after sowing, *M. maydis* caused apparent decreases in vegetative growth parameters with differences between the inoculation methods.

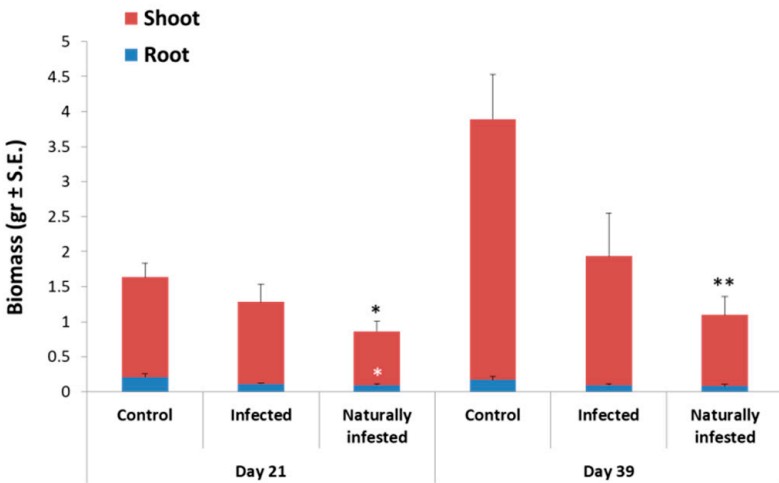

**Figure 8.** Emerging seedlings pathogenicity assay. The late-wilt-sensitive maize Jubilee cv. germination and first development were evaluated by adding culture disks to commercial disease-free soil mixture, 6 days from sowing (infected) or using naturally infested soil taken from a commercial maize field. The same maize plants grown on a commercial disease-free soil mixture without inoculation served as the control group. The biomass of roots or aboveground parts (wet weight) was determined 21 and 39 days post-sowing. Vertical upper bars represent the standard error of the mean of four replications (pots, each containing one plant). When existing, significance from the control is indicated as * = $p <$ 0.05 or ** = $p <$ 0.005.

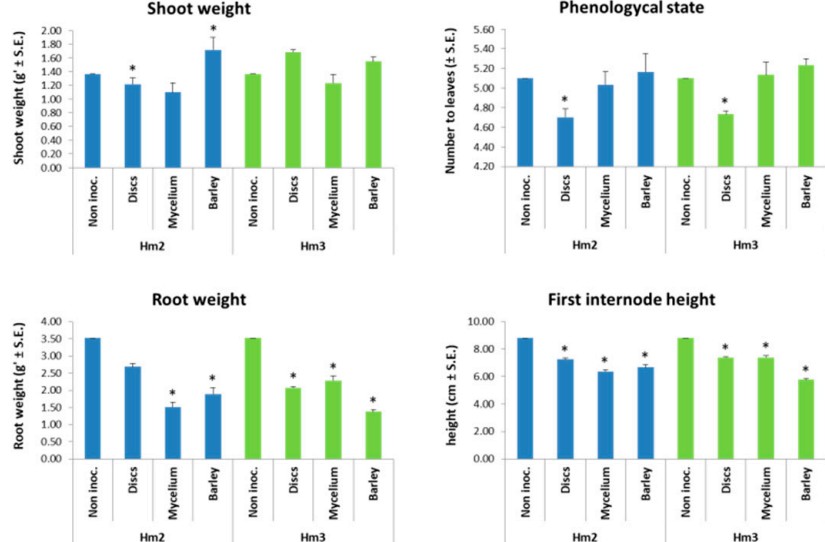

**Figure 9.** Soil inoculation methods to induce late wilt symptoms in the sensitive Prelude cv. sprouts. Three methods were compared here as a *Magnaporthiopsis maydis* complementary inoculum to naturally infested soil: (1) Discs—adding four culture disks (6 mm in diameter) to each seed; (2) Mycelium—adding 200 mg hyphae suspension to each seed; and (3) Barley—adding sterilized inoculated barley seeds to the soil (mixing the top 20 cm of the soil with 20 g of those seeds). The direct application of culture disks or hyphae was made 7–10 days from sowing, when the plants first emerge above the ground surface. The infected sterilized barley seeds soil inoculation was done immediately before seeding. The control group includes non-inoculated plants. Symptoms were estimated 30 days after sowing. Each treatment included six independent replications (pots). Bars indicate standard error. Asterisks indicate significant difference from the control ($p < 0.05$).

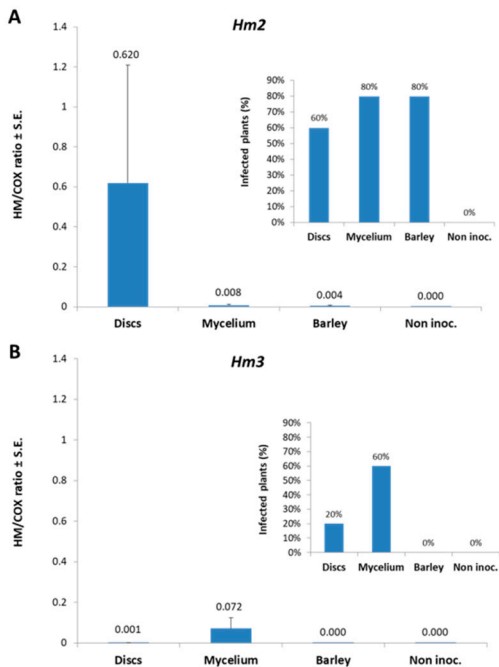

**Figure 10.** qPCR molecular diagnosis of the seedling assay. Real-time qPCR analysis of *Magnaporthiopsis maydis* unique oligonucleotide in maize plant samples of the sensitive Prelude cv. The experimental procedure and the plants' symptoms are described in Figure 9. The two *M. maydis* isolates studied here are *Hm*2 (**A**) and *Hm*3 (**B**). The y-axis indicates *M. maydis* relative DNA (*HM*) abundance normalized to cytochrome c oxidase (*COX*). The insert presents the percentage of infected plants (containing *M. maydis* DNA inside their tissues) identified by the qPCR molecular tracking. Controls: uninfected plants (Non. inoc.). Values indicate an average of five replicates. Standard errors are indicated in error bars.

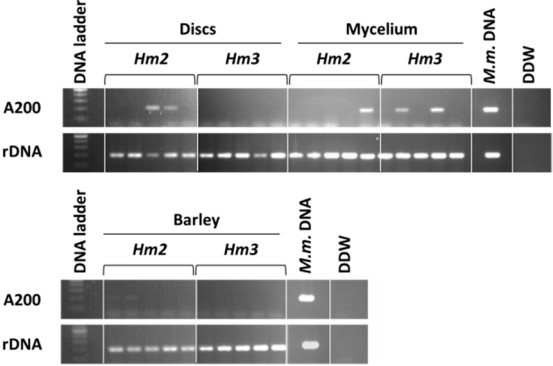

**Figure 11.** Electrophoresis results showing PCR amplification of the seedling assay. The experimental procedure and the plants' symptoms are described in Figure 9. A200—the unique *Magnaporthiopsis maydis* oligonucleotide (**upper panel**). rDNA—18S eukaryotic ribosomal DNA (**lower panel**). The other PCR abbreviations are described in Figure 3.

Infecting the plants by adding colony agar disks or mycelia suspension to the plant roots was found to be more effective than dispersing sterilized barley seeds infected with a pathogen in the soil. According to the symptoms estimated 30 days after sowing (Figure 9), the colony agar disk infection method caused the greatest decrease in the number of leaves (phenological condition) of the maize host plants. The direct application of hyphae suspension (fungal inoculation) to the seeds had a maximum effect on the weight of both roots and aboveground parts, as well as on the height of the first internode. Tracking the pathogen DNA inside the treated plants revealed that in the *Hm2*-infected plants, the colony agar disk method resulted in the highest DNA levels compared to the other inoculation methods and the control (Figure 10). Two isolates of the pathogen were

inspected in this experiment (*Hm2* and *Hm3*). The symptoms analysis carried out at the end of the experiment did not show that one of the two isolates examined was more violent than the other. This is in agreement with the results of the assay inspecting the effect of *M. maydis* culture filtrate on seed germination (Figure 7). However, an analysis of the qPCR results shows that the number of plants detected with *Hm3* DNA was small compared to the plants infected with *Hm2* isolates, and that the overall DNA levels in plant tissues obtained in the treatment with the *Hm3* isolate were lower (Figure 10). Both the PCR and the qPCR had equivalent results (Figures 10 and 11), although the qPCR method was found to be far more sensitive than the traditional PCR method, and enabled a high level of detail between treatments. Yet, as was previously demonstrated [10], due to the presence of samples in which there was no infection (which is commonly found in potted plants and even more so in field experiments), relatively large standard errors occurred that made it difficult to obtain statistical significance (Figure 10). These variations could also be observed in the scattered pattern of the DNA bands within the PCR treatments repetition results (Figure 11).

*Infested field trial.* The qPCR is an important diagnostic research tool that could apply in field experiments to track pathogenesis and evaluate antifungal control treatments. To this end, we conducted a field experiment in a maize field near Kibbutz Amir in the Hula Valley (Upper Galilee, northern Israel) during the spring and summer of 2016. The two representative sweet corn hybrids studied in this experiment included the susceptible Prelude cv. cultivar and the relatively resistant Royalty cv. cultivar. Plant samples were collected at 10-day intervals, enabling the isolation of the DNA from each sample for later diagnosis using qPCR. The time intervals analysis of *M. maydis* pathogenesis in the host tissues using the qPCR method first identified the pathogen 20 days after seeding in the roots of the susceptible Prelude cv. (Figure 12), 10–30 days before conventional PCR detection [9] and the usual appearance of the disease's earliest symptoms (Figure 13). The unique *M. maydis* segment detection percentage (of both maize cultivars combined together) gradually increased from 53% (on day 20) to 89% (on day 30), and to 100% (from day 40 onwards).

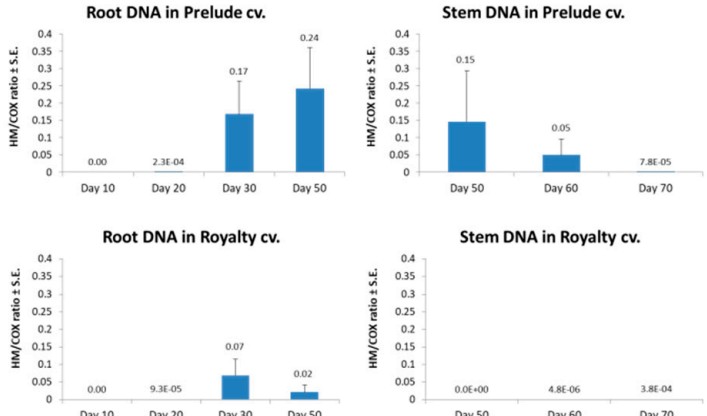

**Figure 12.** qPCR diagnosis of late wilt pathogenesis in the field. The experiment was conducted during the spring and summer of 2016 in an infested field in the Hula Valley in the Upper Galilee in northern Israel. Two representative sweet maize cultivars were evaluated in this experiment: the late-wilt-sensitive Prelude cv. and the relatively resistant Royalty cv. Seeding was performed on 25 May 2016 and germination (with a frontal irrigation system) one day later. Plants emerged above the ground surface approximately six days after planting. Plants were first pollinated when they reached 70% silk on 14 July 2016 (50 days after sowing). Plants were collected arbitrarily at 10-day intervals from sowing onwards. DNA was extracted from the root or aboveground parts of each plant and analyzed in three independent replications. qPCR was performed to amplify a specific *Magnaporthiopsis maydis* segment as described in Figure 10. Bars indicate the mean of three replications of each plant group. Standard errors are indicated in error bars.

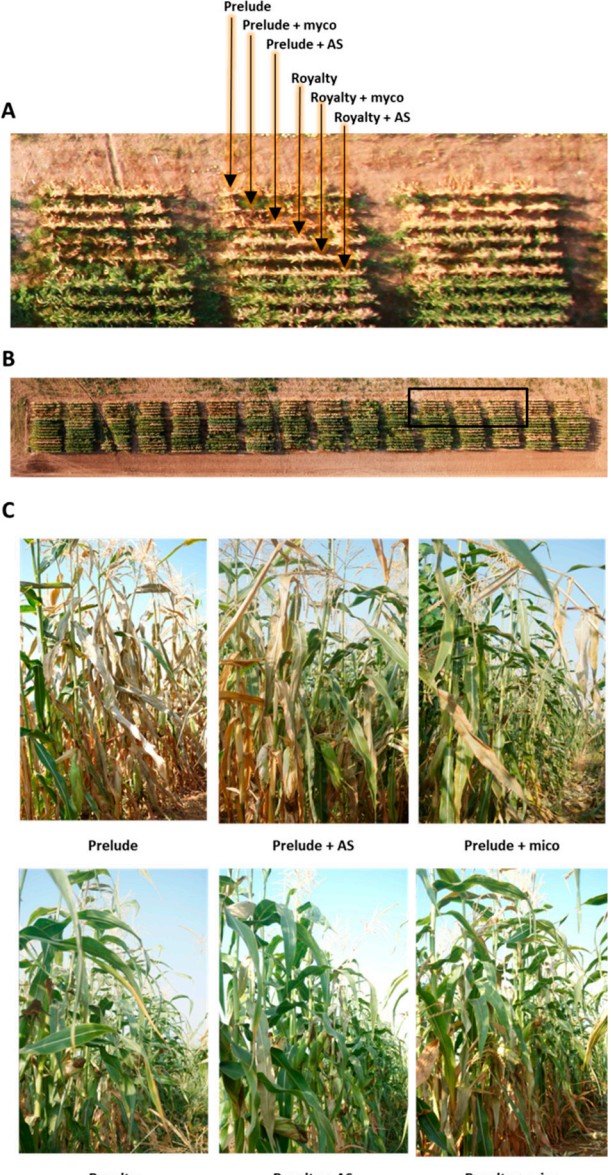

**Figure 13.** Photographs of the experimental field (described in Figure 12). The experimental maize field was photographed from the air by Asaf Solomon 87 days after sowing (37 days after fertilization). (**A**) Close-up of a portion of the aerial photograph. (**B**) The whole field in which the portion shown in (**A**) is marked by a black box. Representative lines of the experiment groups—the control (untreated), the mycorrhizal fungi (mico) seed coating, and the Azoxystrobin fungicide (AS) seed coating—are marked by arrows. At that age, all the plants were dehydrated regardless of the treatment or the maize cultivar. (**C**) Representative wilted Prelude cv. plants (upper panel) and apparently healthy Royalty cv. plants (lower panel) photographed 71 days after sowing (21 days after fertilization). Wilt symptoms include drying out ascending upwards in the plant, including stem and leaf yellowing and dehydration.

Progression of disease symptoms in the Prelude cv. plants followed quantitative changes in fungal DNA estimated by the qPCR analysis. The fungal DNA isolated from the Prelude cv. gradually increased, peaking in the root and stem on 50 DAS (Figure 12). At approximately this time, the plants were fertilized, and early signs of the disease started to appear. Later, from day 50 onwards, the fungal DNA in the stem decreased until it reached a near-zero level on day 70 post-sowing. At approximately this age, 20 days after pollination, most of the Prelude cv. plants (60%) were already diseased and had dried out (Figure 13, Table 2). Interestingly, in the relatively resistant maize Royalty cv., the

fungus DNA was clearly identified in the roots of the plant from day 20 onwards, peaked in the root on 30 DAS, and was close to the discovery threshold in the stem for all measurements (Figure 12). This cultivar showed markedly fewer signs of disease (4% dehydration at 69 DAS, Figure 13, Table 2). The qPCR-based method's sensitivity enables us to detect changes of 40,000 times in DNA levels in maize tissues.

**Table 2.** The efficiency of the Azoxystrobin or mycorrhizal fungi seed coatings in the field.

| Assessment | Days after Sowing | Prelude cv. | | | Royalty cv. | | |
|---|---|---|---|---|---|---|---|
| | | NT [3] | AS | Myco | NT | AS | Myco |
| Root Wet Weight (gr ± S.E.) | 10 | 0.53 ± 0.16 | 0.62 ± 0.12 | 0.58 ± 0.03 | 0.62 ± 0.06 | 0.52 ± 0.09 | 0.51 ± 0.07 |
| Shoot Wet Weight (gr ± S.E.) | 10 | 1.77 ± 0.10 | 2.44 ± 0.29 [4] | 1.93 ± 0.23 | 2.57 ± 0.15 | 2.35 ± 0.56 | 2.06 ± 0.33 |
| Wilting (%) [1] | 69 | 60 | 48.6 | 40.3 [4] | 3.7 | 1.2 | 1.7 |
| | 75 | 100 | 96.3 | 97.1 | 26.7 | 41.9 [4] | 9.3 |
| Yield (kg/m$^2$) [2] | 77 | Less than 0.5 | Less than 0.5 | Less than 0.5 | 1.7 ± 0.10 | 1.7 ± 0.13 | 1.6 ± 0.10 |

[1] Dehydration assessment was done 19 and 25 days after fertilization in sensitive maize Prelude cv. and in the relatively late-wilt-resistant maize Royalty cv. Wilt determination was done by calculating the percentage of plants showing typical late wilt symptoms: color alternation of the upper leaves to light-silver and then to light brown, and rolling inward from the edges of the entire leaf. Results represent the average of six replications. [2] Yield assessment was done 27 days after fertilization included all the upper part plant cobs in a 5-m-long section of each of the experimental rows. On harvest day the sensitive Prelude cv. plots were collapsed and nearly a total yield loss was recorded. [3] NT—control—untreated plots. AS—Azoxystrobin; and Myco— mycorrhizal fungi—plots with chemical and biological seed coating respectively. [4] Significance from the control at $p \leq 0.05$. The calculation was done using Student's *t*-test.

As previously reported [9,23], *M. maydis* slowly spread during the first five weeks after maize germination. When growing susceptible maize cultivars, such as Jubilee, in infested commercial fields, the first symptoms of dehydration usually appeared 50–60 DAS, just before the tasseling stage. The disease symptoms recorded by us in the Prelude cv. were similar to data in the literature. By anthesis (flowering at 9–10 weeks), the lower leaves had turned a light-green to light-silver, then lost their color and became light brown and dehydrated. The dehydration symptoms spread upwards in the infected sensitive maize plants and eventually (75 DAS, 25 DAF), in severe cases (100% in the Prelude cv. and 27% in the Royalty cv.), killed the host (Figure 13, Table 2). Interestingly, two weeks later (87 DAF, 37 DAF), all the plants included in this experiment (the Prelude cv. and the Royalty cv.) dried out (Figure 13A,B), implying that the relative resistance of the Royalty cv. is only a temporary state and eventually this cultivar will collapse. A sharp increase in wilting symptoms in this cultivar in just six days (from 4% to 27%) between 69 and 75 DAS supports this observation. In the Prelude cv. plots, the cobs were poorly developed, and a nearly total crop yield loss was recorded (Table 2). In comparison, in the Royalty cv. plots, a yield of 1.7 kg/m$^2$ was recorded.

A dedicated evaluation was made at 30 DAS in the root and at 60 DAS in the stem to examine the potential of the qPCR-DNA-tracking method in studying the response of the field plants to the preventive treatments (Figure 14). The treatments included Azoxystrobin seed coating and seeds pretreated with mycorrhizal fungi (Rootella S™ product from Groundwork BioAg, Mazor, Israel). For comparison, a conventional PCR was conducted. Compared to the qPCR method, the conventional PCR resultant electrophoretic gel provides similar but less-accurate and less-sensitive results (Figure 14). For example, both methods recorded similar variations in *M. maydis* fungal DNA levels at 30 DAS, with no apparent beneficial effect from any of the seed treatments, but a noticeable reduction as a result of the Azoxystrobin or mycorrhizal fungi seed coating in the Prelude cv. in the stem at 60 DAS (Figure 14B). However, the qPCR method identified a reduction in *M. maydis* DNA in the Prelude cv. on day 60 post-sowing (in the stem) compared to day 30 (in the root), which was not evident in the PCR results. The high sensitivity of the qPCR-based method detected changes of 260,000 times in the DNA levels in maize tissues. This is important when tiny amounts of DNA are measured, as in the

Royalty cv. treatments in the stem at 60 DAS. At that measuring point, the PCR method was unable to detect any fungal DNA in the samples (Figure 14B).

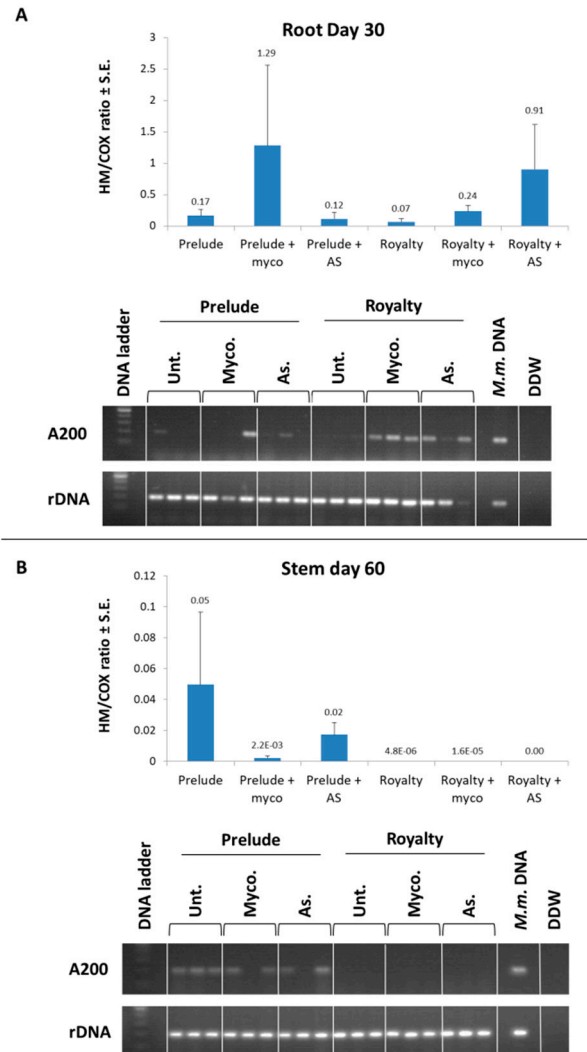

**Figure 14.** Effect of biological and chemical seed coating in the field. Molecular diagnosis of late wilt in maize plant samples in the field experiments described in Figure 12. Two maize cultivars were inspected in this in-depth molecular evaluation: the susceptible cultivar of sweet corn Prelude cv. and the relatively resistant Royalty cv. cultivar. (**A**) Root samples (30 days after sowing) and (**B**) stem samples (60 days after sowing) were inspected for the presence of the pathogen using qPCR (upper panel) or PCR (lower panel) amplification of the unique *Magnaporthiopsis maydis* oligonucleotide. In the qPCR, the y-axis indicates *M. maydis* relative DNA (*HM*) abundance normalized to cytochrome c oxidase (*COX*). Vertical upper bars represent the standard error of the mean of three representative plants sampled arbitrarily from each cultivar. The PCR abbreviations are described in Figure 3.

The chemical and biological seed coatings managed to rescue some of the late wilt symptoms in the sensitive Prelude cv. In sprouts (age 10 DAS), the biomass levels of both the Azoxystrobin -treated plant's root and shoot were higher (17% and 38%, respectively, Table 2). In these sprouts, the mycorrhizal coating resulted in a 9% elevation in biomass levels of both the root and the shoot. In the Prelude plots at approximately 70 DAS, the chemical and biological seed coatings reduced the wilting percentage by 19% and 33%, respectively (Table 2). However, at the end of the growth session, at 75 DAS, the Azoxystrobin and mycorrhizal coatings failed to protect the sensitive Prelude cv. plants and only a minor decrease (4% and 3%, respectively) in dehydration was recorded compared to the

untreated control plants (Table 2). The yield assessment conducted at 77 DAS (27 DAF) was correlated to the severe dehydration symptoms in the Prelude plots, with nearly 100% yield loss. However, in the Royalty plots, regardless of the apparent benefit to the seed coating protection against the disease wilting symptoms, especially in the mycorrhizal treatment (75 DAS), yield production was similar in the seed coating treatments compared to the control (Table 2).

## 4. Discussion

Late wilt disease is considered to be most harmful in commercial maize fields in Israel and a major threat to corn in Egypt, India, and Spain. The disease is gradually continuing to spread to new countries. Average reports (since 2008) of late wilt from new countries is about one country every two-three years. To date, control of the disease had been based on identifying resistant maize hybrids, but, in many cases, these cultivars do not have the highest market value. The fungus can also undergo pathogenic variations, making this method unreliable in the long term. This work presents a thorough review of the use of new methods for detecting, isolating, and studying *M. maydis* and its interactions with the maize host plant under control and field conditions. Each of these aspects was detailed and demonstrated by presenting new results.

The new findings presented in this work include the isolation of *M. maydis* from infested soils and plant tissues using Hygromycin-embedded media plates, and the identification of the pathogen using molecular techniques (PCR and qPCR). Isolation of the pathogen from natural environments, such as soil or plant tissues, in which other relatively fast-growing fungi (such as *Fusarium* spp.) are present can be challenging. Indeed, careful identification of the pathogen is needed. During the 1990s in Israel, it was assumed that the phenomenon of wilting of maize plants in commercial fields was the result of *Fusarium verticillioides* infestation since this pathogen was the most abundant in the infested plant samples. It was eventually proven by Koch's postulates that the direct cause of the wilting is *M. maydis*, while *F. verticillioides* is a secondary invader or opportunist that developed in these attenuated maize plants [9].

The molecular diagnostic is a sensitive and precise method for final confirmation of pathogen identity. Here, we demonstrated the potential use of a PCR molecular assay, which had been proven earlier to be species-specific [5,17], with some modifications in primer selection. However, in the sprouts pathogenicity assay, this method resulted in a very weak electrophoresis gel band (close to the method sensitivity threshold), so a more sensitive and accurate technique was needed. Indeed, we had introduced an already proven qPCR-based method [10], which clearly answered this need. Recently, the use of real-time PCR targeting the ITS region was demonstrated for the same purpose [53].

We also identified the pathogen using more traditional microscopic and colony morphology characteristics. The latter includes spore development and release, and sclerotium body maturation. This is important because the microscopic phenotypes of *M. maydis* described in the literature are very limited (the few examples include [7,9,26,50]). Additionally, spore germination and the detached root assays were presented. These are important techniques for studying *M. maydis* behavior under challenging environmental conditions, as previously demonstrated [29,49]. Selected important plant phenological stages (seed germination, sprout flourish and maturity, and cob production) were chosen to evaluate crucial pathogenicity aspects. The pathogen's impact on the maize host can be assessed by measuring its ability to invade the inner tissues of seeds [32], influence their germination rate [50], inhibit the seedlings thrived [24], and eventually disrupt normal cob production. Each of these stages can be studied individually, or some could be used together to evaluate a particular intervention treatment, as previously demonstrated [10].

Another important issue is mapping the *M. maydis* population and identifying pathogenic variations within the population, and the relationship between fungal varieties. Studies targeting this aim were carried out in Egypt [16] and Spain [18]. Indeed, it was shown that different competitive abilities exist within the *M. maydis* population in Egypt and that pathogenic diversity in Spain enabled the determination of the levels of aggressiveness of this pathogen. These research data play a crucial rule in the decisions and risk assessment that are made before starting a commercial maize field

growth session. As a preliminary step to address this and rapidly screen *M. maydis* field isolates, the fungal culture filtrate seed germination assay, is an efficient way, as previously reported [50] and illustrated here.

All of these means are important for the study and development of new ways to restrict the constant emerging late wilt disease. A scientific plan focused on developing disease control cannot be established entirely on field experiments during the growing season due to the high demand in terms of resources, time, and labor involved in such experiments. Moreover, the lengthy period of time until results are received and the variations in environmental settings lead to inconsistency in the results. Demonstrating this is the field assay presented here for assessing biological and chemical seed coatings in two representative sweet maize cultivars: the sensitive Prelude cv. and the relatively resistant Royalty cv. As previously reported [10], the chemical seed coating alone cannot protect sensitive maize cultivars grown in heavily infested fields against disease outbreak. Instead, this treatment can provide an additional layer of protection when other treatments are applied, and its advantage appears in less-severe cases or when semi-tolerant maize cultivars are sown. It appears that the biological seed coating using mycorrhizal fungi may have the same benefit with the additional advantage of improving nutrition supply (the original aim of this seed coating according to the manufacturer). This should be examined in a future study that would combine the biological seed coating with other treatments, as was done lately for chemical seed coatings [28].

Additional important methods available for studying host resistance and *M. maydis* virulence behavior, had been reported over the years by other researchers. These include injecting spore suspension into the lower stalk of maize breeding/germplasm lines to assess their responses to late wilt (a prerequisite for identifying stable sources of resistance) [54–56], chemical and histological approaches to identify differences between susceptible and resistant maize cultivars [57], the examination of the influence of maize root exudates embedded media on fungal radial growth in Petri dishes for the same goal, monitoring *M. maydis* infection by measuring canopy temperature and the crop water stress index [19], tracking the appearance of necrotic lesions on the roots [21,22], injecting sensitive mature maize plants with the *M. maydis* culture filtrate (secreted metabolites), evaluating stem tissue color alternation and water conductivity to scale the pathogen isolates' aggressiveness [50], and more. As detailed in the Introduction, extended research efforts had been made to evaluate various ways of controlling the disease.

The development of new research tools to study maize late wilt is an ongoing activity. Today, maize late wilt disease and its causal agent *M. maydis* are considered to be exotic and unfamiliar in most parts of the world. Current research on this disease is being led by only a few researchers throughout the world. The destructive potential of late wilt and its gradual spreading are encouraging expanding the research and seeking new ways to control it. Future research may include mapping the pathogen population and pathogenicity variations among them, studying the genetic sources of maize cultivars' resistance, studying the interactions of *M. maydis* with other microorganisms in the soil and inside the host tissues, screening for alternative hosts (other than maize and lupine, cotton, watermelon and *Setaria viridis* [22,33,34]), and developing new and environmentally friendly strategies to prevent disease outbreak and spread.

**Author Contributions:** Conceptualization, O.D., S.D., D.M., and O.R.; methodology, O.D., S.D., D.M., and O.R.; validation, O.D., S.D., D.M., and O.R.; formal analysis, O.D., S.D., D.M., and O.R.; investigation, O.D., S.D., D.M., and O.R.; resources, O.D. and O.R.; data curation, O.D., S.D., D.M., and O.R.; writing—original draft preparation, O.D.; writing—review and editing, O.D., S.D., D.M., and O.R.; visualization, O.D.; supervision, O.D.; project administration, O.D.; funding acquisition, O.D. and O.R., please turn to the CRediT taxonomy for the term explanation.

**Funding:** This research was funded by the Israel Ministry of Agriculture and Rural Development, the Chief Scientist, grant number 21-35-0003, and by the Israel Organization of Crops and Vegetables, Ministry of Agriculture and Rural Development.

**Conflicts of Interest:** The authors declare no conflicts of interest.

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
