# Peer review of "Methods for Studying Magnaporthiopsis maydis, the Maize Late Wilt Causal Agent"

_agronomy, doi:10.3390/agronomy9040181_

Round 1
Reviewer 1 Report
Comments for the authors
General Comments
The paper “Methods for Studying Harpophora maydis, the Maize Late Wilt Causal Agent” describes and comments an extense range of techniques for detecting, identifying and evaluating the virulence of the fungus causing the late wilt disease in different maize varieties. The authors use a wide variety of experiments, and describe them thoroughly. The results are provided in detail, and the discussion is clear in analysing what are the advances and the difficulties in combating this important crop disease.
The article is very well written, and although being extensive and very descriptive, it is clear and reads well.
Suggestions:
Title: I believe that the most recent nomenclature is Magnaporthiopsis maydis.
Klaubauf S, Tharreau D, Fournier E, Groenewald JZ, Crous PW, De Vries RP, Lebrun M-H (2014) Resolving the polyphyletic nature of Pyricularia (Pyriculariaceae). Studies in Mycology 79: 85-120.doi:10.1016/j.simyco.2014.09.004
Line 267: Replace "another means" by "another way of"
Line 502: Maybe better than "accepted" would be "visualized"
Line 806: Replace "genic sources by "Genetic sources"
Please revise species names of reference 53
Author Response
We thank the reviewer for his/her helpful and essential suggestions and corrections. We believe that this revision has improved the manuscript.
Klaubauf S, Tharreau D, Fournier E, Groenewald JZ, Crous PW, De Vries RP, Lebrun M-H (2014) Resolving the polyphyletic nature of Pyricularia (Pyriculariaceae). Studies in Mycology 79: 85-120.doi:10.1016/j.simyco.2014.09.004
The reviewer is correct.
Gams (2000) introduced the genus Harpophora, based on H. radiciola for a group of species that are phialophora-like in morphology, with cylindrical, curved conidia. The ITS phylogeny generated by Ward and Bateman (1999) and Yuan et al. (2010) showed that species of Harpophora were close to or grouped with Gaeumannomyces Arx & D.L. Olivier. Saleh and Leslie (2004) also reported that H. maydis belonged in the Gaeumannomyces-Harpophora species complex, based on ITS, b-tublin and histone H3 gene sequences. Based on a two-locus phylogeny (LSU, RPB1), Klaubauf et al. (2014) recently transferred Harpophora maydis to the genus Magnaporthiopsis J. Luo & N. Zhang and treated Harpophora zeicola as a later synonym of H. radicicola.
The title was corrected and now reads: “Methods for Studying Magnaporthiopsis maydis, the Maize Late Wilt Causal Agent”.
The above paragraph was added to the introduction.
Line 267: Replace "another means" by "another way of"
Corrected as per the reviewer advice.
Line 502: Maybe better than "accepted" would be "visualized"
The word “accepted” was replaced as suggested.
Line 806: Replace "genic sources by "Genetic sources"
Corrected as suggested by the reviewer.
Please revise species names of reference 53
The species names of reference 53 were verified and corrected.

Reviewer 2 Report
I found the manuscript well organized and documented and useful for the advancement of knowledge on Harpophora maydis, the maize Late wilt causal agent, and the methods used in its study. Therefore, after considering the suggestions in the text, I believe that the manuscript can be published on Agronomy with minor revision.
Suggested Revisions:
Discussion section: I suggest to remove the references to figures from the Discussion, since are not normally required here.
Line 736: I suggest to remove this sentence from here and possibly include it in the Introduction
Author Response
We thank the reviewer for his/her helpful and essential suggestions and corrections. We believe that this revision has improved the manuscript.
Discussion section: I suggest removing the references to figures from the Discussion, since are not normally required here.
All the references to figures were removed from the Discussion section, as suggested by the reviewer.
Line 736: I suggest to remove this sentence from here and possibly include it in the Introduction
The sentence was removed from the Discussion section as suggested.
